# Ophthalmic and Genetic Features of Bardet Biedl Syndrome in a German Cohort

**DOI:** 10.3390/genes13071218

**Published:** 2022-07-08

**Authors:** Fadi Nasser, Susanne Kohl, Anne Kurtenbach, Melanie Kempf, Saskia Biskup, Theresia Zuleger, Tobias B. Haack, Nicole Weisschuh, Katarina Stingl, Eberhart Zrenner

**Affiliations:** 1Centre for Ophthalmology, University of Tübingen, 72076 Tuebingen, Germany; susanne.kohl@med.uni-tuebingen.de (S.K.); anne.kurtenbach@uni-tuebingen.de (A.K.); melanie.kempf@med.uni-tuebingen.de (M.K.); nicole.weisschuh@uni-tuebingen.de (N.W.); katarina.stingl@med.uni-tuebingen.de (K.S.); ezrenner@uni-tuebingen.de (E.Z.); 2Department of Ophthalmology, University of Leipzig, 04103 Leipzig, Germany; 3Center for Rare Eye Diseases, University of Tübingen, 72076 Tuebingen, Germany; 4Praxis für Humangenetik, 72076 Tuebingen, Germany; saskia.biskup@humangenetik-tuebingen.de; 5Institute of Medical Genetics and Applied Genomics, University of Tübingen, 72076 Tuebingen, Germany; theresia.zuleger@med.uni-tuebingen.de (T.Z.); tobias.haack@med.uni-tuebingen.de (T.B.H.); 6Werner Reichardt Centre for Integrative Neuroscience (CIN), University of Tübingen, 72076 Tuebingen, Germany

**Keywords:** BBS, genotype, phenotype, ophthalmology

## Abstract

The aim of this study was to characterize the ophthalmic and genetic features of Bardet Biedl (BBS) syndrome in a cohort of patients from a German specialized ophthalmic care center. Sixty-one patients, aged 5–56 years, underwent a detailed ophthalmic examination including visual acuity and color vision testing, electroretinography (ERG), visually evoked potential recording (VEP), fundus examination, and spectral domain optical coherence tomography (SD-OCT). Adaptive optics flood illumination ophthalmoscopy was performed in five patients. All patients had received diagnostic genetic testing and were selected upon the presence of apparent biallelic variants in known BBS-associated genes. All patients had retinal dystrophy with morphologic changes of the retina. Visual acuity decreased from ~0.2 (decimal) at age 5 to blindness 0 at 50 years. Visual field examination could be performed in only half of the patients and showed a concentric constriction with remaining islands of function in the periphery. ERG recordings were mostly extinguished whereas VEP recordings were reduced in about half of the patients. The cohort of patients showed 51 different likely biallelic mutations—of which 11 are novel—in 12 different BBS-associated genes. The most common associated genes were *BBS10* (32.8%) and *BBS1* (24.6%), and by far the most commonly observed variants were *BBS10* c.271dup;p.C91Lfs*5 (21 alleles) and *BBS1* c.1169T>G;p.M390R (18 alleles). The phenotype associated with the different BBS-associated genes and genotypes in our cohort is heterogeneous, with diverse features without genotype–phenotype correlation. The results confirm and expand our knowledge of this rare disease.

## 1. Introduction

Over 100 years ago, Georges Bardet first published a report on “Congenital obesity syndrome with polydactyly and retinitis pigmentosa (RP)” [1]. Two years later, in 1922, Artur Biedl reported on a “Sibling pair with adiposo-genital dystrophy, along with illustrations of a third case” [2]. Both papers describe a syndrome that was subsequently named the Bardet–Biedl syndrome (BBS, OMIM 2099000), a form of the similar manifesting Laurence–Moon syndrome [3], although there is research suggesting that the two conditions are not distinct, but variable expressions of the same disease [4,5,6,7]. Therefore, the disorder is sometimes also acknowledged as Laurence–Moon–Bardet–Biedl syndrome.

BBS is a rare autosomal recessive genetic disorder belonging to the ciliopathies. It has a prevalence of around 1 in 125,000–160,000 in Europe [8,9]; however, this is considerably higher in closed communities such as the Faroe islands at 1:3700 [10], Bedouin communities of Kuwait at 1:65,000 [11] and the island of Newfoundland at 1:17,500 [12]. 

BBS has a complex phenotype and heterogeneous genotype. It can affect multiple organ systems and has a wide variability of phenotypic expression. Beales, in his notable survey, described 109 patients and their families in the UK [5], and lists five main features of the syndrome: rod-cone dystrophy, polydactyly, short stature along with obesity, learning difficulties and renal tract abnormalities. Moreover, there is a wide range of other medical conditions which may be present including neuropsychiatric abnormalities, colonic disorders, gallstone disease and asthma, and there is considerable inter- and intra-familial variation in phenotype [7,9,12,13]. The Laurence–Moon syndrome may differ from BBS by the presence of spasticity and the absence of polydactyly and obesity [4]. Some symptoms of BBS generally are not present at birth but appear gradually, and progressively worsen during or after the first decade of life, making the average age of diagnosis relatively late at 9 years [5]. Although the combination of symptoms in BBS is variable, a retinal dystrophy develops in almost all patients [5]. 

More than 20 BBS- or BBS-like associated genes have been identified to date, all following an autosomal recessive mode of inheritance (https://sph.uth.edu/retnet/, accessed on 17 July 2022) [14,15]. BBS proteins are necessary for the development of many organs [6]. Ansley [16] was the first to show that BBS is caused by a defect at the basal body of ciliated cells, with BBS proteins found in the basal body and cilia of cells. Subsequently, studies on animal models of BBS confirmed the early findings and have clarified the primary role of the BBS proteins in mediating and regulating intraflagellar transport, a microtubule-based intracellular transport process [17,18,19,20,21,22,23]. Seven of the gene products assemble together with the protein BBIP1/BBIP10 into the BBSome, an octameric protein complex localized at the basal body and involved in trafficking of cargo to and from primary cilia [24,25]. Therefore, BBS is considered to belong to the ciliopathies.

After the initial descriptions of BBS, several case reports on this extremely rare disease were published [26,27,28,29,30,31]. However, over the past 30 years not only has there been a vast increase in the number of genes identified, but also in the number of cases reported [5,8,9,12,32,33]. This is due to development of next-generation sequencing genetic technologies, which has accelerated the identification of the genetic causes of the disease including effective diagnostic genetic testing. In this study, we report on 61 patients that had attended the specialized out-patients’ clinic for inherited retinal dystrophies (IRD) of the University of Tübingen Eye Hospital for diagnosis and therapy. 

## 2. Materials and Methods

### 2.1. Patients

Sixty-one BBS patients with likely biallelic mutations in any known BBS-associated gene and a clinical diagnosis of BBS were included in the study, performed at the Centre for Ophthalmology, University Tübingen, a German specialized ophthalmic care center for inherited retinal diseases. The minimum clinical criteria to establish a clinical diagnosis of BBS were at least four major features (i.e., visual disorder, limb defect, small stature/overweight, learning difficulties, and renal tract abnormalities) or three major and two minor features (e.g., developmental delay, neurological and motor defect, behavioral abnormality, speech and/or hearing deficits, dental anomaly, asthma, facial features, hypogonadism, heart defect, and diabetes mellitus), as suggested by Beales and co-workers in 1999 [5]. All patients received diagnostic genetic testing, either by IRD, BBS or allied disease gene panel sequencing or virtual BBS panel analysis based on whole-genome sequencing.

The patients in this study were selected upon the presence of (likely) biallelic mutations in known BBS-associated genes and upon the diagnosis of BBS based on at least four major features or three major and two minor features [5]. In the initially selected cohort, there were five patients, however, who were non-syndromic and thus these were subsequently not included in this analysis.

The study was approved by the ethics committee of the University of Tübingen. It was carried out in accordance with the Declaration of Helsinki. Written consent was obtained from all patients (or their parents/guardians if underaged or intellectually disabled) for both the research study and for the diagnostic genetic testing.

### 2.2. Procedure

The medical history of the patients was recorded, and a comprehensive ophthalmic examination was performed including: full-field electroretinography (ERG, scotopic and photopic) and visually evoked potential recording (VEPs) according to ISCEV standards with the Diagnosys system (Lowell, MA, USA) or the RETeval^®^ device (LKC Technologies, Inc., Gaithersburg, MD, USA). Fundus autofluorescence (AF, 30° or 55°) and spectral domain optical coherence tomography (SD-OCT) were performed with the Spectralis HRA+OCT (Heidelberg Engineering Inc., Heidelberg, Germany). Adaptive optics flood illumination ophthalmoscopy (AO-FIO) images were additionally obtained from five patients using the adaptive optics flood illuminated camera rtx1^TM^ (Imagine Eyes, Orsay, France). Visual acuity was tested with Snellen or Lea charts, and color vision testing was carried out with the panel D-15 or Lea tests. Additionally, visual field testing was performed with the Octopus 900 (Haag-Streit, Wedel, Germany) using the III4e or I4e stimulus. Of note, some patients with accompanying disabilities were not able to perform all tests. The number of patients that performed each examination is given in the results section. 

## 3. Results

Sixty-one patients from 57 families with likely biallelic mutations in known BBS-associated genes were included in the study (see Appendix A). In Appendix A, a detailed description of the ophthalmological findings for each patient is given, while Appendix A summarizes all systemic findings. 

The patient’s mean age at the time of the ophthalmic examination was 24.5 years +/− 12.3 SD (median 23 years), with 35 males and 26 females ranging from 5 years to 56 years. Figure 1A depicts the age distribution for each gender. In Figure 1B, the visual acuity for both eyes are plotted against age. Visual acuity is extremely compromised at all ages and decreases by approximately 0.06 (decimal) per year. The best visual acuities recorded were 0.63 in a 21-year-old patient with *BBS**12*, 0.5 in a *BBS**2* patient aged 8 years and 0.4 in a *BBS**16* patient aged 42 years.

We evaluated the spherical equivalent in diopter (D) of the patient’s refractive error (Table 1), and the number of patients with myopia and hyperopia for each genetic subgroup are provided separately (of note, nine patients could not perform this test). A myopia grade −3D–−6D is the most common along with a hyperopia <+3D. The patient numbers for the other genes are too low to draw conclusions, but interestingly, the three patients in the subgroups *BBS3, BBS4* and *BBS5* are myopic. The values for the left eye were similar to those of the right (Appendix A). 

The cohort showed 51 different disease-causing variants in twelve different BBS genes, as depicted in Figure 1C,D: Twenty patients (32.8%) carried likely biallelic (apparent homozygous or two heterozygous) variants in the *BBS10* gene and 15 (24.6%) in the *BBS1* gene, five patients (8.2%) in the *BBS9* gene, three patients (4.9%) each in the genes *BBS2*, *BBS3*, *BBS4*, *BBS7*, *BBS5* and *BBS12*, whereas only one patient each (1.6%) carried apparent homozygous disease-causing variants in the *BBS6*, *BBS8*, and *BBS16* genes.

The most common associated genes were *BBS10* (32.8%) and *BBS1* (24.6%), and by far the most commonly observed variants were *BBS10* c.271dup;p.C91Lfs*5 (21 alleles) and *BBS1* c.1169T>G;p.M390R (18 alleles) (see Appendix A). In contrast, 45% of all variants were observed only once or twice. Thirty-nine patients were (apparent) homozygous (63.9%) and 22 patients carried two heterozygous mutations. Segregation to confirm homozygosity or compound-heterozygosity was available in 34.4% (21/61) of the cases. Missense variants were the most common mutation type (47%), followed by frame-shifting small insertion, deletion and duplication mutations (19.6%), variants likely resulting in mis-splicing (13.7%) and nonsense mutations (7.8%). Large deletions were observed in the *BBS1* and the *BBS9* gene. 

In 12 patients, additional heterozygous variants in other BBS- or IRD-related genes were observed (Appendix A). It has been implied that the mutational load or digenic, triallelic variants contribute to the BBS phenotype [34,35]. We have classified the variants (Appendix A), and present only variants that are either classified as variants of uncertain significance, likely pathogenic or pathogenic. Whether these variants contribute to the disease-phenotype could not be established. Of note, both brothers of family BBS56 were homozygous both for the *BBS5* variant c.143-4_143-2ins400-500;p.? and the *BBS12* missense variant of uncertain significance c.1139C>T;p.T380I. Segregation analysis showed that the unaffected mother is also homozygous for the *BBS12* missense variant c.1139C>T;p.T380I; therefore, we conclude that the BBS phenotype is primarily caused by *BBS5* variant c.143-4_143-2ins400-500;p.?, and the *BBS12* variant might—if at all—only modulate the phenotype.

In Figure 2 and Figure 3, we show the retinal images obtained from representative patients. Two patients did not perform the OCT examination and 18 the AF.

The fundus, AF and OCT images of one patient (BBS83) harboring disease-causing variants in the *BBS10* gene are shown in the second row in Figure 2. For comparison, normal images can be seen in the upper row. Peripapillary atrophy, retinal pigment epithelial (RPE) changes with macular atrophy and attenuated vessels, typical of RP, can be seen with a normal optic disc; 14 of the 20 patients with variants in the *BBS10* gene showed a pallor optic disc. A perimacular hyperfluorescent ring can be seen in the AF images (center) which was found in 13 of the 20 patients. Hypoautofluorescent spots are found in the periphery. The OCT images of all *BBS10* patients showed atrophy of the photoreceptor layer and loss of photoreceptor cells, mostly along with internal limiting membrane wrinkling. 

All 15 patients with mutations in the *BBS1* gene show a large macular atrophy with diffuse RPE atrophy throughout the retina and attenuated vessels (third row, Figure 2). Pallor of the optic nerve can also be seen; four of the 15 patients had a normal optic disc. In the AF images, a perimacular hyperfluorescent ring is present, as found in all but three of the 15 patients. All patients also exhibited hypoautofluorescent spots. The OCT images show again foveal atrophy or loss of the photoreceptor layer in all patients. ONL thinning is evident in the *BBS1* patient (BBS30) in the third row in Figure 2. 

In the lower panel in Figure 2, we show the retinal images of one of the five patients carrying disease-causing variants in the *BBS9* gene (BBS42-I). Loss of the photoreceptor layer, foveal atrophy and a pallor optic disc can be seen, as in the patients with mutations in the *BBS10* or *BBS1* genes. The *BBS9* patient also exhibits a bull’s eye maculopathy, seen as a dark macula surrounded by paler rings in the AF images, in addition to hypoautofluorescent spots. This was also found in another two patients with this variant. The OCT displays a loss of the photoreceptor cell layer and foveal atrophy. Two of the patients were brothers (BBS42-I, BBS42-II).

Thus, for the three largest subgroups of patients *BBS10* (20 patients), *BBS1* (15 patients) and *BBS9* (five patients) there is a similar pattern of retinal degeneration. This also holds true for the other genes present in our patient cohort, as demonstrated in Figure 3, where there are only three or fewer patients in each subgroup. Of interest is that the fundus images of patients BBS44-I (*BBS3*) and BBS58 (*BBS12*) show evidence of cone dystrophy. The patient with the *BBS12* mutations shows additionally a peripapillary myopic conus.

Although the results of the retinal imaging indicate that the phenotype of the retinal dystrophy does not appear to depend greatly on the mutations present, two features deserve further mentioning. The first is the presence of a bull’s eye macula, observed in nine (15%) of the patients, which although not uncommon in IRD, appears to be unevenly distributed between subgroups. It was found in 3/5 patients with disease-causing variants in the *BBS9* gene and 2/3 patients with variants in the *BBS7* gene, but was not present in any of the 20 patients with disease-causing genotypes in the *BBS10* gene. A bull’s eye was further found in 1/15 *BBS1* patients, 1/3 *BBS2* patients, 1/3 *BBS3* patients and 1/3 *BBS4* patients (see Appendix A). A second feature is the presence of cone-rod dystrophy. A total of 16 patients (26%) were found to have a cone or cone-rod dystrophy, but it appears to be distributed evenly between BBS subtypes (3/15 *BBS1*, 2/3 *BBS2*, 3/3 *BBS3,* 1/5 *BBS9*, 5/20 *BBS10*, and 2/3 *BBS12*).

To explore the photoreceptor abnormalities on a cellular level, adaptive optics imaging was performed in five patients. In at least one eye of each patient, we could acquire a montage out of five single 4 degrees × 4 degrees sized images. The images of three patients are shown in Figure 4. Due to the nystagmus in three of the five patients, the imaging was difficult. However, in all images a disrupted cone photoreceptor mosaic dominated by dark patchy areas was observable. In one patient (BBS10-II), we detected a hemicycle shaped RPE clumping in both eyes, which is also observable in the fundus and infrared imaging. In patient BBS85 (third panel), the AO images showed a typical central blur followed by a reduced but still visible cone photoreceptor mosaic followed by parafoveal blur. The images correlate with the still delineable ellipsoid zone and external limiting membrane in the foveola in the OCT, which disintegrate parafoveally. Furthermore, many small hyperreflective spots were detectable in the AO imaging. In another patient (BBS75), the AO imaging revealed the appearance of so-called “puffy cones”; these cones appear bigger in size but in a still intact mosaic. The fixation of patient BBS75 was too unstable to overlay the AO imaging with the fundus autofluorescence or the infrared imaging.

### 3.1. Further Examinations

In all patients, the first symptoms were night blindness and/or photophobia along with eye abnormalities such as strabismus, cataracts, and nystagmus (Appendix A). Twenty of the 61 patients (33%) suffered from strabismus with two having had it as a child and two having had surgery. Thirty-two of the 61 patients have or have had a cataract, nine of whom had received surgery, but they were distributed evenly between genotypes. Fourteen (70%) of the 20 *BBS10* patients suffered from nystagmus, and 4/5 (80%) of the *BBS9* patients but only 4/15 (27%) patients with *BBS1*. 

Visual field testing could not be performed in 35 of the patients. Of the rest, most showed a concentric constriction, with some showing good function or remaining islands of function in the periphery. The ERG was extinguished in 54 of the 61 patients; five patients (two *BBS12* (BBS58, 16 y and BBS59, 21 y), one *BBS10* (RCD768 29 y), two *BBS3* (BBS44-I, 23 y and BBS44-I, 13 y) showed a scotopic response. Recording was not possible in two patients. The VEP was performed in 24 subjects (four pattern VEP, 20 flash VEP). Thirteen showed a good flash VEP response and a reduction in VEP amplitude was found in 10 patients. Color vision testing was not possible in nine patients and only five gave normal results. Forty-four patients were totally color blind.

### 3.2. Systemic Features 

In the Appendix A, the systemic features for each patient are listed. An evaluation of primary and secondary features of the cohort is given in Table 2, showing that all patients suffered from retinal dystrophy, with polydactyly and obesity also being common. Brachy-/syndactyly is the most common (55%) among the secondary features. None of our patients reported a hearing impairment and only one patient reported anosmia.

## 4. Discussion

In this study, we report on the ophthalmic and genetic features of a cohort of 61 BBS patients, ranging from 5 to 56 years in age. The initial symptoms reported by almost all of our patients were night blindness and photophobia with only seven patients reporting additional amblyopia or visual acuity impairment.

The visual impairment of the BBS patients is severe. The visual acuity of the BBS cohort decreased from an average of around 0.2 in young children to 0 at age 50 (Figure 1). This is comparable to that reported elsewhere [14,36,37,38,39]. All of the patients showed changes in the morphology of the fundus in the fundus photographs and AF images, demonstrating that retinal dystrophy is a general deficiency in BBS patients [14,36,37,38,39]. RP along with “general imperfections of development” were the first features to be reported about the syndrome [3]. The retinal dystrophy occurring after early childhood can be explained by a theory that proposes that the dendritic processes of retinal neurons are supported by intraflagellar transport protein complexes [40,41] which are responsible for the assembly and replacement of cilia [42], as well as signaling in the cilium [43]. Mutations in the BBS genes disturb ciliary assembly (see e.g., [44,45,46]). The degeneration was also evident in the reduced cone and rod responses in the ERG of most patients. We find extinguished responses in 89% of the patients, aged between 13 and 29 years, which is consistent with the results of Fawcett et al. [47], indicating that there is no correlation between age and the degree of retinal dysfunction revealed by electrophysiological recordings. The VEP response, on the other hand, was reduced in about half of the recordings. These results are in agreement with previous studies [47,48,49,50]. The visual field could not be measured accurately in around half of the patients due to inadequate fixation and nystagmus. Most of the remainder showed a central scotoma with peripheral islands of remaining function. 

Observations of the cone photoreceptor mosaic with an AO flood-illuminated camera or an AO-SLO in retinitis pigmentosa patients have shown that the cone density can range from normal to severely reduced [51,52,53,54,55,56]. Additionally, four main patterns could be described, which could be correlated to the progressive phases of retinal degeneration [54]. The AO images of our BBS patients complement the standard clinical examinations and the findings of the OCT and fundus photography. The disrupted cone photoreceptor mosaic, dark patchy areas and clumping of the RPE demonstrate the severe dystrophy of the retina. A publication of two siblings with BBS due to a mutation in the *BBS7* gene also observed a reduced photoreceptor density by adaptive optics scanning light ophthalmoscopy (AOSLO) [57]. Detection of changes in cone appearance in one patient could also show the further progression of degeneration. 

EURO-WABB, a rare disease registry, has published guidelines for the management of BBS and the assessment of the many affected organs and systems (www.euro-wabb.org, accessed on 14 February 2022). Generally, our patients have similar features (Table 2) to those in the guidelines, but the frequencies with which we find them differ. Thirty-one percent of our patients show renal abnormalities, whereas in the guidelines this figure is only 9%. We also find a larger number of patients with a speech disorder (25% vs. 2%), developmental delay (34% vs. 9%) and brachydactyly/syndactyly (55% vs. 4%) and a lower number of patients with heart disease (0% vs. 6%). The Laurence–Moon syndrome generally differs from BBS by the presence of spasticity and the absence of polydactyly and obesity [4], which was not observed in any of our patients.

The results of five non-syndromic patients were not included in this analysis. Most interesting is the 20-year-old brother of patient BBS40-I, aged 43, who only displayed retinal symptoms of the disease despite his brother showing developmental delay, mental retardation and polydactyly. The brothers were both homozygous for the *BBS9* c.263+1G>T;p.(?) variant. In line with previous studies, 3/5 non-syndromic patients carried disease-causing variants in the *BBS1* gene, all homozygous for *BBS1* c.1169T>G;p.(M390R) [58,59]. The fifth patient who appeared non-syndromic was heterozygous for two variants in *BBS4* c.883C>T;p.(R295*) and c.1107-10_-7delTCTG;p.(?). Non-syndromic retinitis pigmentosa has also been reported in patients with other BBS mutations [60,61,62]. 

*BBS1* and *BBS10* were the most common genotypes in our cohort, as found by others [9,13,63], together making up 57.4% of our patients. Mutations in *BBS9* were our next most common cause of BBS, at 8% (five patients). Other studies, on the other hand, have reported significant proportions of *BBS2* and *BBS12*-related disease [63,64]. Ethnic factors including common founder mutations and rate of consanguinity are expected to be the cause of this discrepancy.

The mutation spectrum is comparable to that of other recent studies [35,65,66], and 51 different variants were observed, either (apparent) homozygous or two (compound) heterozygous variants found in 12 different genes (Appendix A). Of these, 11 variants have never been reported before and deserve further discussion. In *BBS1*, four novel variants were observed; one near splice site c.479+4A>G;p.(?), a duplication of 10 bp c.784_793dup;p.(N269Gfs*95) and a 17 bp deletion c.1431_1447del;p.(L478Rfs*17), both resulting in frame-shift and premature termination codon, as well as a large deletion covering exons 14 to 17. While the latter three are predicted pathogenic or likely pathogenic according to ACMG classification, the splice site variant c.479+4A>G;p.(?) is classified as variant of uncertain significance. Of note, the variant is predicted to result in mis-splicing by two of three queried prediction software and we therefore suggest this compound-heterozygous variant in patient BBS78 to be likely disease-causing together with the deletion of exons 14–17 on the *BBS1* counter allele.

In the *BBS5* gene, an intronic insertion of 400–500 bp was observed in three patients of two independent families. Both families have documented consanguinity, but it could not be established whether these patients are distantly related. Unfortunately, the exact extent and sequence content of the insertion could not be established due to repetitive sequences, and therefore also prediction on the effect of this insertion for example on splicing could not be predicted. In addition, segregation analysis showed that the unaffected mother of both brothers of family BBS56 was homozygous for the variant. Still, we cannot exclude that this *BBS5* insertion may contribute to the disease, e.g., as a hypomorphic allele, as the variant may alter the splice acceptor of exon 3, was observed in multiple patients and families, and was absent from healthy control individuals (i.e., gnomAD browser database). 

Another putative splicing variant was observed in the *BBS7* gene. The variant c.1037+29T>A;p.(?) was found in patient BBS60, compound heterozygous to the 4 bp deletion c.712_715delAGAG;p.(R238Efs*59). It is not predicted to result in mis-splicing but was the only rare heterozygous variant found in this patient by BBS panel sequencing. We cannot exclude that another *BBS7* variant, for example, a deep intronic variant, was missed and only cDNA analysis or minigene splice assays could finally elucidate the effect of this variant. 

The *TTC8* (BBS8) missense variant c.694G>A;p.(G232R) was found apparent homozygously in patient BBS67 and was classified as a variant of uncertain significance. No further support for its pathogenicity than the output of the prediction tools, which include consideration to conservation and biophysical properties of the amino acid residue, and frequency of the variant in normal population, can be added.

Last but not least, three new variants were observed in *BBS10*, which is the most frequently mutated gene in our study. The missense variants c.686C>T;p.(P229L), c.901C>T;p.(L301V) and c.1802C>T;p.(P601L) also fulfil the same criteria as those just described for the novel *TTC8* missense variants to classify these as variants of uncertain significance. Further cases, segregation analysis and functional studies will be needed to confirm the pathogenic effect of these variants. In contrast, the *BBS10* 2 bp duplication c.858_859dup;p.(Q287Lfs*12) will result in frameshift and premature termination codon, resulting in loss of important structural and functional domains and rendering this variant possibly a null allele.

We find that the phenotype of the different BBS genes in our cohort is heterogeneous, with diverse features occurring in all mutations. This is characteristic of the syndrome [6,38,67]; however, in recent years, by analyzing larger patient cohorts, some associations have been published. 

Generally, *BBS10* and *BBS2* patients have been reported as having more severe features than *BBS1* [63,68,69] with lower risk of cardiovascular disease [70]. We did find a significant difference between BBS gene-association with respect to the occurrence of nystagmus (*p* = 0.005), with 68% of *BBS10* patients suffering from nystagmus and only 21% of those with *BBS1*. Our *BBS10* patients also had a greater chance of strabismus 37% (compared to the *BBS1* 21%). This difference between BBS10 and BBS1 also tended to be true for the systemic symptoms, but the lack of sufficient numbers does not allow a more detailed analysis. In addition, renal anomalies have been found to be more prevalent in *BBS2*, *BBS7* and *BBS9* patients [63,68], and a relatively low penetrance of polydactyly in patients with mutations in *BBS1*, which does not appear to be the case in our patients.

The visual prognosis for BBS patients is poor. However, genetic therapeutics offer a more promising future, with research ongoing in animal models. In Bbs1 and Bbs4 mouse models, the use of gene therapy to preserve the retinal function, especially at early stages of the disease, has been shown to be successful [71,72,73]. Clinical trials using antisense-oligonucleotide therapy for the common deep intronic variant c.2991+1655A->G in *CEP290* (BBS14/NPHP6/LCA10/MKS4/SLSN6/JBTS5) have shown promising results in halting degeneration of the photoreceptors, although the recent press release by ProQR on the phase II/III trial is unfortunately less encouraging as primary endpoints were not met [74]. The olfactory system has also been shown to be sensitive to gene therapy in mice [75,76]. A macaque model of RP due to mutations in *BBS7* has been recently discovered which will help in testing treatments for this subtype [77] and further the search for adequate therapies.

## 5. Conclusions 

The results of the 61 BBS patients in this German cohort, confirm and expand our knowledge on the phenotype and genotype of BBS and will aid in providing information for future studies to help combat the devastating effects of this rare disease.

## Figures and Tables

**Figure 1 genes-13-01218-f001:**
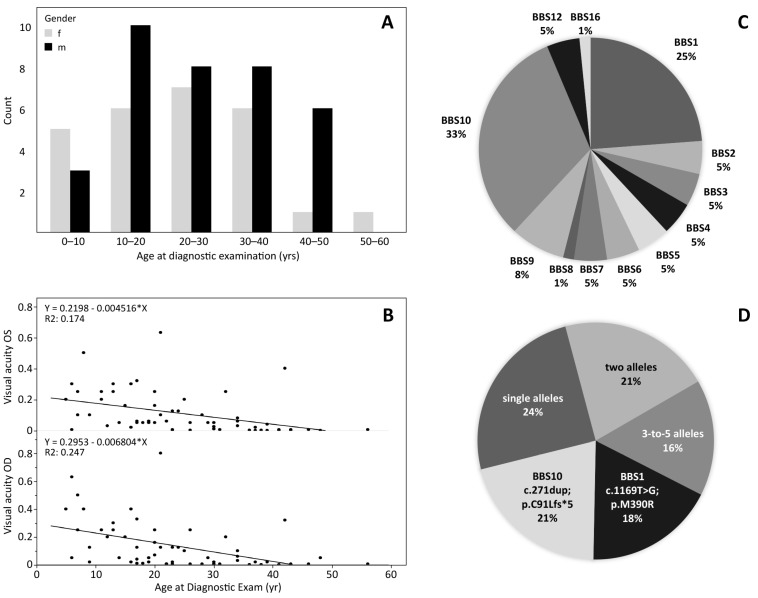
Demographics of the 61 BBS patients. (**A**): Age of the patients, for male and female separately. (**B**): Decimal visual acuity of the left and right eyes plotted against age. (**C**): Frequency of BBS genes carrying biallelic mutations in the patients of this study. (**D**): Common variants presented as percentage of total allele counts. *BBS10* c.271dup;p.C91Lfs*5 (21%) and *BBS1* c.1169T>G;p.M390R (18%) are most common, but also other variants with up to 5 alleles were observed recurrently.

**Figure 2 genes-13-01218-f002:**
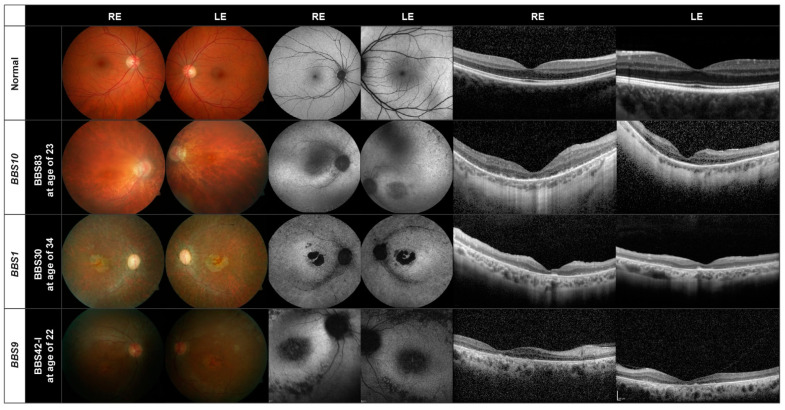
Fundus (**left**), AF (**center**) and OCT (**right**) images of representative patients with disease-causing variants in the *BBS1, BBS10,* and the *BBS9* genes. The images of a normal eye are shown in the first row.

**Figure 3 genes-13-01218-f003:**
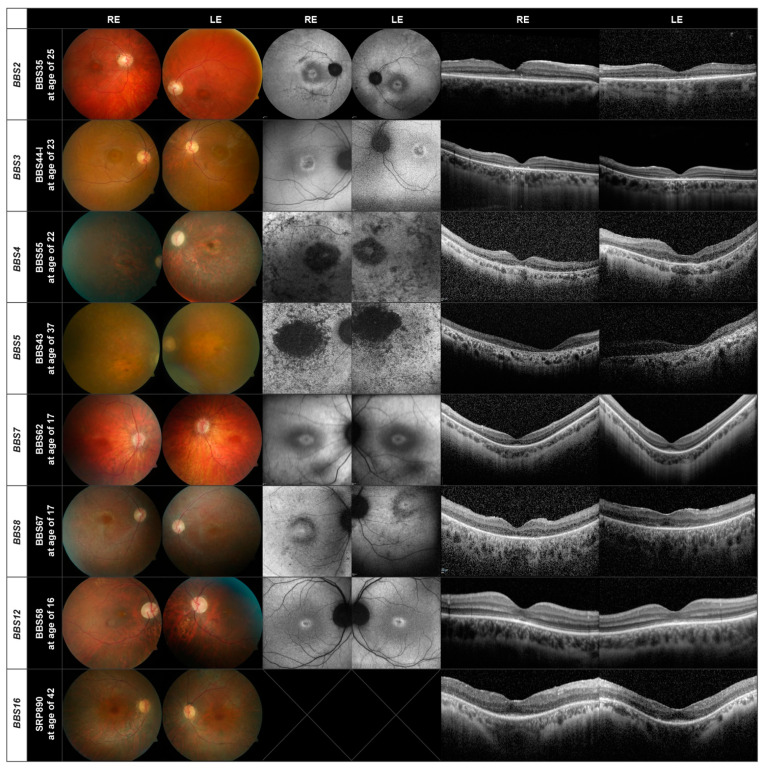
Fundus, AF and OCT images of one patient each carrying disease-causing variants in the *BBS2, BBS3, BBS4*, *BBS5, BBS7, BBS8, BBS12*, and *BBS16* genes.

**Figure 4 genes-13-01218-f004:**
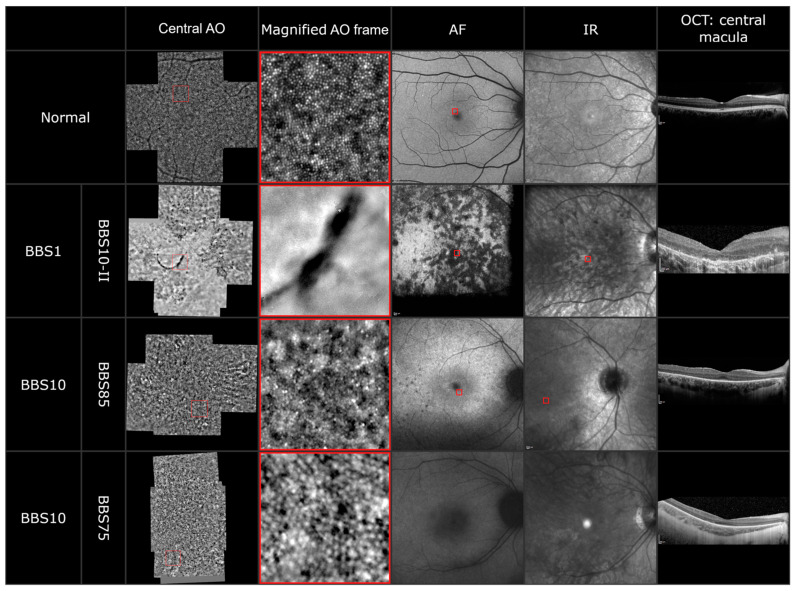
AO montage, magnified AO imaging (highlighted by a red frame), AF, IR and OCT of a healthy subject and three BBS patients carrying disease-causing variants in the *BBS1* and *BBS10* genes.

**Table 1 genes-13-01218-t001:** Spherical equivalent (dpt) of the right eye. Number of patients and % of cases for each genetic subgroup.

Type	Myopia		Hyperopia	
	−3D–−6D	>−6D	<+3D	≥+3D
*BBS1* (10)	1 (10%)	1 (10%)	3 (30%)	2 (20%)
*BBS2* (3)	2 (67%)		1 (33%)	
*BBS3* (3)	1 (33%)	2 (67%)		
*BBS4* (3)	1 (25%)	2 (67%)		
*BBS5* (2)	1 (50%)			
*BBS6* (1)		1(100%)		
*BBS7* (3)	1 (33%)		1 (33%)	1 (33%)
*BBS8* (1)				1(100%)
*BBS9* (5)	2 (50%)		2 50%)	
*BBS10* (17)	7 (41%)		4 (24%)	1 (6%)
*BBS12* (3)	2 (67%)		1 (33%)	
*BBS16* (1)			1(100%)	

**Table 2 genes-13-01218-t002:** Primary and secondary features of the BBS study cohort.

Primary Features	%	Secondary Features	%
Retinal dystrophy	100	Brachy-/syndactyly	55
Polydactyly	87	Developmental delay	34
hands and feet	43	Speech delay	25
hands	14	Diabetes mellitus (Type 2)	8
feet	30	Hirschsprung disease	8
Obesity	79	Liver/gall bladder disorders	2
Renal abnormality	31		
Learning difficulties	28		
Genital abnormality	10		

## Data Availability

Not applicable.

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
