# Peer review of "Ophthalmic and Genetic Features of Bardet Biedl Syndrome in a German Cohort"

_genes, 2022, doi:10.3390/genes13071218_

Round 1
Reviewer 1 Report
The manuscript entitled “Ophthalmic and genetic features of Bardet Biedl syndrome in a German cohort” by Fadi Nasser et al. reviews the ophthalmic status of 61 patients diagnosed with BBS. It is an interesting and important study.
Comments
1. Resolution in Fig. 1B is very low. Parts of Fig 1A is also low. The graph in 1B is fuzzy and hard to read. The fuzziness seam related to the graph-images and not to added text. Axes legends, symbols and lines in 1B are fuzzy and cannot be read. There is problem with some of the small image-related text.
2. Figures 2- 4 is on the boarder of being fuzzy and has too low resolution. Particularly the white text on black background is hard to read. Should be improved.
3. A table with affected and BBS associated genes and the BBS-type, would improve the paper and assist the reader. The table may include genes, mutations, gene functions and molecular functions (and ie. GO classification) as well as appropriate references.
4. Line 210: “Bull’s eye macula” should be explained and pointed out in the figure(s).
5. Line 386-387. Meaning of the sentence unclear: “.. three novel variants were observed in BBS10, the most commonly mutated in gene.”
Author Response
- Resolution in Fig. 1B is very low. Parts of Fig 1A is also low. The graph in 1B is fuzzy and hard to read. The fuzziness seam related to the graph-images and not to added text. Axes legends, symbols and lines in 1B are fuzzy and cannot be read. There is problem with some of the small image-related text.
Reply: We have reformatted all Figures.
- Figures 2- 4 is on the boarder of being fuzzy and has too low resolution. Particularly the white text on black background is hard to read. Should be improved.
Reply: As above.
- A table with affected and BBS associated genes and the BBS-type, would improve the paper and assist the reader. The table may include genes, mutations, gene functions and molecular functions (and ie. GO classification) as well as appropriate references.
Reply: We provide 5 Supplementary Tables (unfortunately not originally uploaded), the first 3 of which list the information requested.
- Line 210: “Bull’s eye macula” should be explained and pointed out in the figure(s).
We have rephrased to “The BBS9 patient also exhibits a bull`s eye maculopathy, seen as a dark macula surrounded by paler rings in the AF images, in addition to hypoautofluorescent spots. This was also found in another two patients with this variant.”
- Line 386-387. Meaning of the sentence unclear: “.. three novel variants were observed in BBS10, the most commonly mutated in gene.”
Reply: Now lines 389-390 We have rephrased the sentence to: “three novel variants were observed in BBS10, which is the most frequently mutated gene in our study.”
Reviewer 2 Report
An excellent and succinct presentation of a large group of patients with BBS. It offers an extensive description of the associated eye condition with its variations and genetic background. Provides significant reference body and great illustrations.
Author Response
Thank you for the positive comment on our manuscript.
Reviewer 3 Report
In this genetic landscape manuscript, the authors described an extensive cohort of patients with BBS with a highlight in the eye phenotype.
The manuscript, in general, is well structured but needs improvements and corrections. In addition, I could not consult the supplementary data. Was it uploaded?
In the introduction, it is unclear when the authors are talking about BBS diagnosis or the diagnosis of BBS ophthalmological presentation. Please, make it clear what diagnosis is being described.
Methods: Should be more clear the context of the center that follows these patients. Are the authors working at an Ophthalmological center? In that case, most patients would have an eye phenotype that elicited the referral, or was not that the case? This could be a topic for the discussion.
Details should be given about the type of genetic testing performed and the classification used for pathogenicity.
line 134: Mutations in the two most commonly mutated genes BBS1 and BBS10 are more likely to be associated with myopia than hyperopia.
Given the small sample, it is challenging to accept the description of an association. Frequency data is enough.
line 175: Of note, both brothers of family BBS56 were homozygous both for the BBS5 variant c.143-4_143-2ins400-500;p.? and the BBS12 missense variant of uncertain significance c.1139C>T;p.T380I. Segregation analysis showed that the unaffected mother is also homozygous for the BBS12 missense variant c.1139C>T;p.T380I, therefore we conclude that the BBS phenotype is primarily caused by BBS5 variant c.143-4_143-2ins400-500;p.?, and the BBS12 variant might – if at all – only modulate the phenotype. What was the ACMG classification of these variants after the segregation studies? If the BBS12 variant is not relevant, why is it described here?
line 266: 4% -> 40%
line 266: the p-value was related to what comparison? Did the data have a parametric distribution to use the T-test? The statistical analysis and software used should be described in the Methods section.
line 275: The majority were totally colour blind. The number of patients with the phenotype should be included.
line 335: In the initially selected cohort, there were five patients, however, who were non-syndromic and thus these were subsequently not included in this analysis. Include this selection in the methods, not in the discussion.
Figure 1, 2, and 3 are blurred. It is not possible to read the text in 1A and 1B.
Figure 1D it hard to interpret. It needs additional description in the figure legend. Correct typo: single allele.
Author Response
The manuscript, in general, is well structured but needs improvements and corrections. In addition, I could not consult the supplementary data. Was it uploaded?
Reply: Sorry, the supplementary data are now accessible.
In the introduction, it is unclear when the authors are talking about BBS diagnosis or the diagnosis of BBS ophthalmological presentation. Please, make it clear what diagnosis is being described.
Reply: We have deleted that sentence, as it appears to be redundant.
Methods: Should be more clear the context of the center that follows these patients. Are the authors working at an Ophthalmological center? In that case, most patients would have an eye phenotype that elicited the referral, or was not that the case? This could be a topic for the discussion.
Details should be given about the type of genetic testing performed and the classification used for pathogenicity.
Reply: Thank you for this comment. This information has already been provided at the end of the Introduction, but we have now also added the information that we are working in a specialized center for ophthalmology to the Abstract and Material & Methods section.
line 134: Mutations in the two most commonly mutated genes BBS1 and BBS10 are more likely to be associated with myopia than hyperopia.
Given the small sample, it is challenging to accept the description of an association. Frequency data is enough.
Reply: We have removed this sentence.
line 175: Of note, both brothers of family BBS56 were homozygous both for the BBS5 variant c.143-4_143-2ins400-500;p.? and the BBS12 missense variant of uncertain significance c.1139C>T;p.T380I. Segregation analysis showed that the unaffected mother is also homozygous for the BBS12 missense variant c.1139C>T;p.T380I, therefore we conclude that the BBS phenotype is primarily caused by BBS5 variant c.143-4_143-2ins400-500;p.?, and the BBS12 variant might – if at all – only modulate the phenotype. What was the ACMG classification of these variants after the segregation studies? If the BBS12 variant is not relevant, why is it described here?
Reply: The variant would likely be classified as VUS or likely benign. But it is important to understand that hypomorphic alleles and modifiers are known in inherited retinal dystrophies and BBS, and we do feel it is important to keep this information for other experts in this field.
line 266: 4% -> 40%
Reply: This has been corrected
line 266: the p-value was related to what comparison? Did the data have a parametric distribution to use the T-test? The statistical analysis and software used should be described in the Methods section.
Reply: We have deleted this comparison.
line 275: The majority were totally colour blind. The number of patients with the phenotype should be included.
Reply: We have changed this sentence to read “Forty-four patients were totally color blind.”
line 335: In the initially selected cohort, there were five patients, however, who were non-syndromic and thus these were subsequently not included in this analysis. Include this selection in the methods, not in the discussion.
Reply: We have moved this to the Methods section.
Figure 1, 2, and 3 are blurred. It is not possible to read the text in 1A and 1B.
Reply: We have reformatted all Figures.
Figure 1D it hard to interpret. It needs additional description in the figure legend. Correct typo: single allele.
Reply: We have altered the legend to read : “Common variants presented as percentage of total allele counts. BBS10 c.271dup;p.C91Lfs*5 (21 %) and BBS1 c.1169T>G;p.M390R (18 %) are most common, but also other variants with up to 5 alleles were observed recurrently”. The typo has been corrected.